# Relationship between Slip Severity and BMI in Patients with Slipped Capital Femoral Epiphysis Treated with In Situ Screw Fixation

**DOI:** 10.3390/jpm13040604

**Published:** 2023-03-30

**Authors:** Jun-Hyuk Lim, Hyeongmin Song, Gyo Rim Kang, Sungmin Kim, Sung-Taek Jung

**Affiliations:** 1Department of Orthopedic Surgery, Chonnam National University Medical School and Hospital, Dong-gu, Gwangju 61469, Republic of Korea; ove03@naver.com (J.-H.L.); songmh1206@naver.com (H.S.); 2Department of Orthopedic Surgery, Chonnam National University Hospital, Dong-gu, Gwangju 61469, Republic of Korea; gyorim0120@gmail.com

**Keywords:** slipped capital femoral epiphysis, in situ screw fixation, body mass index, slip severity, obesity

## Abstract

**Background**: Slipped capital femoral epiphysis (SCFE) is a hip disorder that occurs in adolescence before epiphyseal plate closure, causing anatomical changes in the femoral head. Obesity is known to be the single most important risk factor for idiopathic slipped capital femoral epiphysis (SCFE), which is highly related to mechanical factors. Meanwhile, as increased slip angle increases major complications in patients with SCFE, slip severity is an important factor to evaluate prognosis. In obese patients with SCFE, higher shear stress is loaded on the joint, which increases the likelihood of slip. The study aim was to assess the patients with SCFE treated with in situ screw fixation according to the degree of the obesity and to find any factors affecting the severity of slip. **Methods**: Overall, 68 patients (74 hips) with SCFE who were treated with in situ fixation screw fixation were included (mean age 11.38, range: 6–16) years. There were 53 males (77.9%) and 15 females (22.1%). Patients were categorized underweight, normal weight, overweight, and obese depending on BMI percentile for age. We determined slip severity of patients using the Southwick angle. The slip severity was defined as mild if the angle difference was less than 30 degrees, moderate if the angle difference was between 30 and 50 degrees, and severe if the angle difference was greater than 50 degrees. To examine the effects of several variables on slip severity, we used a univariable and multivariate regression analysis. The following data were analyzed: age at surgery, sex, BMI, symptom duration before diagnosis (acute, chronic, and acute on chronic), stability, and ability to ambulate at the time of the hospital visit. **Results**: The mean BMI was 25.18 (range: 14.7–33.4) kg/m^2^. There were more patients with overweight and obese than those with normal weight in SCFE (81.1% vs. 18.9%). We did not find significant differences between overall slip severity and degree of obesity or in any subgroup analysis. **Conclusions**: We did not find a relationship between slip severity and degree of obesity. A prospective study related to the mechanical factors affecting the slip severity according to the degree of obesity is needed.

## 1. Introduction

Slipped capital femoral epiphysis (SCFE) is a hip disorder that occurs in adolescence before epiphyseal plate closure, causing anatomical changes in the femoral head [1]. As the femoral neck moves anterolateral, the femoral epiphysis is displaced posteriorly relative to the neck [2]. Incidence of SCFE varies from 0.33 to 24.58 per 100,000 among studies, and the typical age at onset is known to be between 8 and 15 years old. SCFE usually occurs in association with a high body mass index (BMI) around the time of the pubertal growth spurt. Risk factors such as age, sex, ethnicity, BMI, and geographic and seasonal variations have been linked to the development of SCFE in general [3].

The goal of treatment is to prevent further slippage of the femoral epiphysis, and some authors advocate an aggressive approach such as open reduction with slip stabilization. In situ screw fixation is well accepted as the generally preferred treatment for stable slipped capital femoral epiphysis (SCFE). This approach lowers the risk of slip progression, preventing worsening of the deformity and helping avoid the complications associated with unstable slips, including avascular necrosis and chondrolysis.

There are two etiologic classifications of SCFE: atypical and idiopathic. Atypical SCFE includes patients with obvious underlying disorders (e.g., endocrinopathy, renal failure, radiation therapy, etc.) [3]. Although the etiology of SCFE can be extremely complex [4,5], most SCFEs have an idiopathic etiology and are strongly associated with obesity [6,7,8,9,10,11].

Obese pediatric patients experience changes in hip biomechanics during gait as their hip joint reaction force increases in proportion to body weight [12,13]. Mechanical factors are highly associated with SCFE development. Cadaver studies have suggested that increased forces may lead to SCFE in these obese children. In obese patients with SCFE, higher shear stress is loaded on the joint, which increases the likelihood of slip [14,15].

The slip angle of SCFE is an important factor used to evaluate its prognosis, and an increased slip angle increases major complications [16,17]. One study had reported that slip severity is affected by the patient’s age and duration of symptoms [18]. However, studies on the relationship between the degree of obesity and slip severity are limited. In another study, the slip angle was increased in non-obese patients rather than in obese patients, but the difference was not statistically significant [19].

We tried to find out whether the mechanical force applied to the hip could change according to the degree of obesity and whether this was related to the slip severity. To address this dearth in the related literature, this study aimed to evaluate the correlation between slip severity and the degree of obesity in patients who had been treated with in situ screw fixation for SCFE.

## 2. Materials and Methods

The Institutional Review Board of the Chonnam National University Hospital (IRB No. CNUH-2023-033) approved this retrospective study and waived the informed consent requirement from patients due to the retrospective nature of the study. We identified all consecutive patients with a diagnosis of SCFE who had undergone in situ screw fixation between 1 January 2002, and 31 December 2020 at this hospital.

We performed all surgeries with in situ screw fixation regardless of slip severity. Under general anesthesia, the affected leg was held in extension and slight internal rotation position on the traction table in supine position, and hemi-lithotomy position was performed on unaffected leg. After positioning the patient, we checked whether the true anterior to posterior (AP) hip radiograph and lateral radiograph of the affected leg could be checked with the C-arm. Under fluoroscopic guidance, a cannulated screw guidewire was inserted percutaneously in the center-center position of the deformed femoral epiphysis. The guidewire was inserted up to about 3 mm from the femoral articular surface. After making a small stab incision around the guidewire, the bone was drilled. A 6.5 or 7.3 mm partial-threaded cannulated screw was inserted over the guidewire after reaming. After cannulated screw insertion, internal and external rotation was carefully performed on the traction leg to confirm that the screw had not penetrated into the articular surface. The guidewire was removed, and the stab wound was closed after confirming the satisfactory screw position. Patients were allowed protected partial weight bearing with crutches for 6 weeks. We checked serial hip radiographs at 3 months, 6 months, and 1 year to check for any complications on the affected side and newly detected slip on the unaffected side.

We initially identified 80 pediatric patients who were treated with in situ screw fixation for SCFE. We included patients with AP hip radiographs or frog-leg lateral hip radiographs for determination of slip angle. We evaluated data from the time of the initial visit, including symptoms, duration of symptoms, height, and weight. The following patients were excluded from the analysis to ensure that the results were as accurate and reliable as possible, without any potential confounding factors that could affect the outcome: one was excluded because the radiographs taken at the beginning of treatment were not clear enough to analyze, two were excluded because they underwent prophylactic screw fixation on the contralateral hip, two because their surgery was performed at an outside institution, one who had medical comorbidities, two with obvious endocrine dysfunction that was difficult to categorize as idiopathic SCFE, one patient whose was missing height and weight data, and three patients whose duration of symptoms and walking ability at the time of initial visit on the medical chart were unclear.

A total of 68 patients (74 hips) were included in the study. The mean age was 11.38 (range: 6–16) years. There were 53 males (77.9%) and 15 females (22.1%).

The BMI is a value calculated by weight divided by height squared and is an objective and consistent parameter with high sensitivity and specificity that can be successfully employed to categorize obese and non-obese patients for clinical measurement [20,21]. The BMI percentile for age is an indicator that displays height and weight scales depending on age on the same graph. Unlike in adults, BMI percentile for age is recommended for use in pediatric patients aged 2 to 20 years to evaluate overweight or obese patients [22].

Using the method recommended by the Center for Disease Control (CDC), patients were categorized as the following: underweight if their BMI was less than the fifth BMI percentile for age; normal weight if they were in the 5th to 84th BMI percentile for age; overweight if in the 85th to 94th BMI percentile for age; obese if their BMI was greater than the 95th BMI percentile for their age group [23,24].

The following data were obtained: age at surgery, sex, BMI, symptom duration before diagnosis, and ability to ambulate at the time of the hospital visit. Acute slips caused symptoms lasting <3 weeks, whereas chronic slips caused symptoms for a period of at least 3 weeks. Acute on chronic slips caused symptoms of more than 3 weeks but showed remodeling of the femoral neck on radiographic imaging [25,26].

Patients were categorized based on their ability to bear weight at the time of visit. The stable group could manage full weight bearing or partial weight bearing, while the unstable group was unable to bear any weight [26].

The Southwick angle was used to evaluate the slip severity of a unilateral SCFE based on the radiograph at the time of visit. The Southwick angle is measured on the frog-leg lateral radiograph of the hip and is defined as the angle between a line drawn through the center of the femoral neck and a line drawn through the center of the femoral shaft. The severity of SCFE is generally classified based on the Southwick angle. Mild SCFE is defined as a Southwick angle of less than 30 degrees, moderate SCFE is defined as a Southwick angle of 30 to 50 degrees, and severe SCFE is defined as a Southwick angle of greater than 50 degrees. The slip severity was defined as mild if the angle difference was less than 30 degrees, moderate if the angle difference was between 30 and 50 degrees, and severe if the angle difference was greater than 50 degrees [27] (Figure 1A). For patients with bilateral SCFE, we determined slip severity by measuring the Southwick angle, which uses the control angle of 12 degrees as a reference point [28] (Figure 1B). Two authors (H.S. and G.R.K.) who were not involved in the clinical care of the patients participated in the radiographic image measurements.

### Statistical Analysis

All continuous variables were expressed as means with ranges. Data were assessed for normality on plots using the Shapiro–Wilk test. Descriptive statistics were compiled for all variables. To compare continuous variables in all three groups according to the degree of obesity, the Kruskal–Wallis rank test was used. Fisher’s exact test was used to compare the categorical variables. The relationship between slip severity and body mass index percentile for age was analyzed using Pearson correlation coefficients. To examine the effects of several variables on slip severity, we used univariate regression analysis. Statistical analyses were performed using SPSS^®^ version 25.0 software (IBM Corp, Armonk, NY, USA), with the level of significance set at *p* < 0.05.

## 3. Results

The patients’ demographic data are summarized in Table 1. The mean BMI was 25.18 (range: 14.70–33.39). There were no “underweight” patients; 18.9% (14 patients) were considered “normal weight” with BMI percentiles between 5% and 85%, and 13.5% (10 patients) fell within the “overweight” category with BMI percentiles between 85% and 95%. Lastly, 67.6% (50 patients) had BMI percentiles >95% and were categorized as “obese”. Thus, 81.1% of patients were overweight or obese.

We had only three patients with severe slip. There were thirty-five hips with chronic slip.

The data were further analyzed by categorizing the results based on the accepted BMI percentile ranges (Table 2). There were no significant differences between the groups with regard to BMI except the age. The mean age of patients with normal weight was higher than that of patients who were overweight or obese (13.93 vs. 11.1 or 10.72, *p* < 0.001). There were no significant differences in slip severity between the groups with regard to BMI.

Overall, there was no significant correlation between the BMI percentile for age and slip severity overall or in the mild and moderate slip groups (Figure 2A–C).

## 4. Discussion

Obesity is well known as the single most significant risk factor in patients with idiopathic SCFE [6,7,8,10,11,18], and BMI can be used as a risk indicator for SCFE [8]. BMI percentile for age is commonly used to categorize the degree of obesity in pediatric patients. As lean body mass increases in patients with pediatric obesity, the compressive and vertical shear contact force of the hip joint increases, which affects hip joint loading during walking [12]. In particular, the slip severity is an important factor in the prognosis of SCFE, and as the slip angle increases, the possibility of major complications increases [16,17]. We hypothesized that a higher BMI percentile for age has a significant effect on hip biomechanics and could promote slip especially in obese patients with SCFE. We subdivided the degree of obesity depending on the BMI percentile for age and sought to evaluate the relationship between the degree of obesity and slip severity in patients with SCFE.

In our study, we did not find an association between the degree of obesity and slip severity according to the BMI percentile for age (*p* = 0.09). Similarly, several authors discussed the relationship between obesity and slip severity, but they also did not find any relationship between these variables. Obana et al. reported that slip severity seemed to increase more in non-obese patients rather than in obese patients, but this did not show a statistically significant difference [19]. Similarly, Loader et al. did not also find an association between BMI and slip severity [18]. Our results are similar to those of the previous studies in that we did not find an association between BMI and slip severity.

In subgroup analysis, we did not find any positive correlations in the mild slip severity group and moderate slip severity group. Mechanical factors are known to be important for development of SCFE, and excessive weight gain itself can reduce the resistance of the proximal femoral growth plate [15,29]. Obese patients with SCFE show subtle changes in the diameter of the epiphyseal plate that do not increase proportionally with body weight and height, and slip occurs more easily when excessive shear stress is applied to the still-immature and open epiphyseal plate [14,15]. Additionally, other studies have reported that obesity causes anatomical changes in the hip joint, such as decreased femoral anteversion and excessive remodeling of the femoral neck, which leads to epiphyseal failure [28,30]. However, in these studies, the mechanical factor affected according to the degree of obesity was not identified. In our study, as we did not find any correlation in the entire slip group, we cannot confirm that the degree of obesity increases the slip severity. Mechanical studies or a prospective study according to the degree of obesity is needed.

In univariate analysis, we also failed to find any factors affecting slip severity. In a retrospective study of 243 patients with a total of 328 stable SCFEs, Loader et al. reported that the patient’s age and duration of symptoms were associated with slip severity through multivariate analysis. This study also showed that slip severity increased 2.0 times and 4.1 times, respectively, when the age at diagnosis was 12.5 years or older, and the duration of symptoms was 2 months or longer [18]. In our study, age and duration of symptoms were not factors related to slip severity, but further studies affecting slip severity are needed.

We found that the mean age of patients with SCFE was lower in the overweight or obese group and higher in the normal-weight group, which was statistically significant. (*p* < 0.001). In one study, the typical age at onset of SCFE ranged from 8 to 15 years, and the mean age was 12.0 years in male patients and 11.2 years in female patients, with most patients over the 95th percentile [3]. Patients with idiopathic SCFE that developed at a typical age are mostly obese, which is consistent with our results. Several authors discussed the association between age and obesity. Poussa et al. suggested that patients with SCFE already have a high BMI in early years [8], and Song et al., reported that there is a higher association in those less than 10 years of age [31]. In particular, Obana et al. found that patients with SCFE developing at older age tended to be skinnier than those without [19]. Our findings were similar to the previous studies described above.

Our study group was comprised of 57 male patients and 17 female patients, indicating that SCFE is a more common in boys. Other studies have also reported that SCFE is more common in boys, with a male-to-female ratio of about 1.3 to 1 [3]. Although this current study noted a difference in the ratio, it is still similar to the results of previous studies that have reported a more frequent occurrence in male patients. However, our study noted a different proportion of patients with bilateral SCFE compared to other studies. Loader et al. reported that among patients newly diagnosed with SCFE, between 18–50% were diagnosed with bilateral SCFE [4]; our study identified only 13% of patients with the same condition. This difference may be due to the small sample size of our single-center study.

In our study group, the average age of patients was 11.4 years. Loader et al. reported that the average age of SCFE onset is 12 years for males and 11 years for females and that the underlying etiology should be evaluated to differentiate idiopathic SCFE from atypical SCFE [32]. In our study, we routinely performed hormonal tests on all patients diagnosed with SCFE and classified patients with negative results as idiopathic SCFE.

The causes of SCFE are diverse, and atypical SCFE is associated with endocrine dysfunction, chronic renal insufficiency, and growth hormone therapy [33,34,35,36,37]. Recently, Halverson et al. reported that elevated serum leptin levels, aside from BMI, have physiological effects related to the disease state of SCFE. One possible mechanism for this relationship is that leptin may affect the growth plate in the hip joint, which is the area where SCFE occurs. Leptin has been shown to stimulate the growth of osteoblasts and inhibit the activity of osteoclasts, which could lead to an imbalance in bone growth and an increased risk of SCFE. Overall, while the relationship between SCFE and leptin is not fully understood, elevated leptin levels may be a risk factor for SCFE even independent of obesity status [38]. Vitamin D deficiency and abnormal cortical bone density are also given attention as factors related to SCFE in a prospective pilot study. The results showed that the children with SCFE had significantly lower levels of vitamin D and higher levels of parathyroid hormone compared to the controls. The radiographs also showed that the children with SCFE had more severe subclinical rickets compared to the controls. They concluded that subclinical rickets could contribute to the development of SCFE, but further studies are needed to confirm this association. However, more research is needed to establish a definitive link between these two conditions [39]. As SCFE is a disorder caused by multifactorial events, as established by previous studies, it can be assumed that there may be other unknown factors that affect slip angle, which is why further research is needed to delineate important prognostic factors.

Even though obesity is known to affect hip biomechanics and mechanical factors, and therefore, it was expected that there would be a correlation between obesity and slip severity in SCFE, the study did not find such a correlation. This suggests that SCFE is a complex disease that involves multiple factors, and the newly discovered correlations between SCFE and leptin and SCFE and vitamin D are not yet fully understood. It is possible that other unknown factors that affect the slip angle may be involved, and further research is needed to understand these factors better.

Our study has several limitations. First, our study is a retrospective study. Second, due to the rare nature of the disease [40,41,42], the number of patients in the severe slip severity group is relatively small compared to other groups, so there is a possibility that the small sample size limits the statistical power of this study. However, we tried to maintain diversity in SCFE severity by including severe patients. It is considered that future studies with more severe patients are needed. Third, in situ screw fixation was performed for all patients with SCFE regardless of the initial slip severity. This may have introduced selection bias. In particular, only in situ screw fixation was used in severe SCFE patients. The patients with severe SCFE were all chronic SCFE. After sufficiently discussing the treatment with the patient and parents, in situ screw fixation was decided. Fourth, due to the diversity of SCFE causes, patients expected to have atypical SCFE may have been classified as idiopathic SCFE. However, we excluded patients with medical comorbidity, and there were no patients with abnormalities in hormonal tests. Lastly, there is a slight difference in the patient ratio from the existing incidence study of SCFE. Nevertheless, our study is valuable because it is the first study of the relationship between BMI and slip severity in a Korean setting treated at a single institute.

## 5. Conclusions

We found more SCFE patients were overweight and obese than those in the normal BMI range. The mean age of patients with normal weight was higher than that of obese patients. We did not find any relationship between slip severity and degree of obesity. In addition, the subgroup analysis using BMI percentile for age did not find any correlation. A prospective study related to the mechanical factors affecting the slip severity according to the degree of obesity is needed.

## Figures and Tables

**Figure 1 jpm-13-00604-f001:**
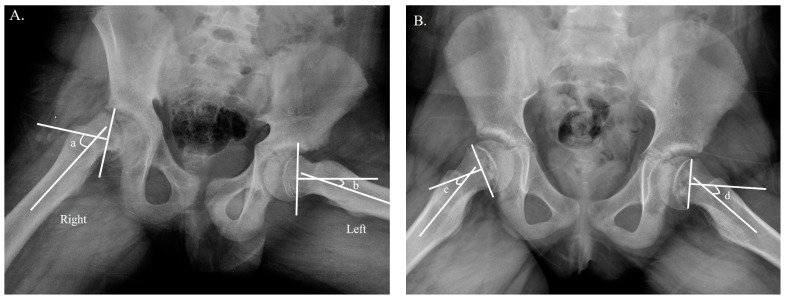
Determination of slip severity. (**A**) Measurement of the Southwick angle difference on a frog-leg lateral radiograph in unilateral SCFE. The slip angle difference was calculated by subtracting b from a; in the example above, the difference was between 30 and 50 degrees, which was classified as moderate slip [27]. (**B**) Measurement of the Southwick angle in bilateral SCFE on a frog-leg lateral radiograph, which uses 12 degrees as the control angle [28]. The slip angle difference was calculated by subtracting c from d; in the example above, both sides were less than 30 degrees, which was classified as mild slip.

**Figure 2 jpm-13-00604-f002:**
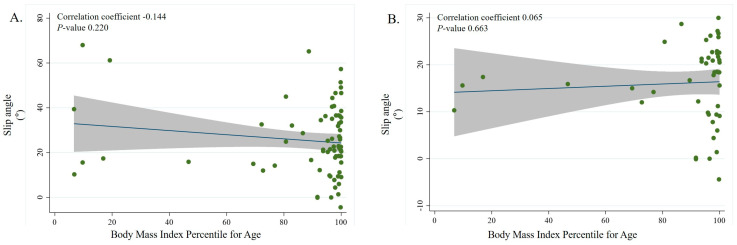
Scatterplots showing the relationship between slip angle represented by the Southwick angle difference in SCFE and BMI percentile for age. In the figure, the blue color line is the linear prediction plot and the gray zone means the 95% confidence interval. Subgroup correlation between BMI percentile for age and slip severity overall (correlation coefficient = −0.144, *p* < 0.220) (**A**), for mild slip severity (correlation coefficient = 0.065, *p* < 0.663) (**B**), and moderate slip severity (correlation coefficient = 0.106, *p* < 0.631) (**C**). We performed univariate analysis to find any variables affecting slip severity (Table 3). However, no variable was found to significantly affect slip severity in univariate analysis.

**Table 1 jpm-13-00604-t001:** Overall patient demographic data.

Measures	Value
Number of patients (hips)	68 (74)
Sex	
Males	53 (77.9%)
Females	15 (22.1%)
Mean Age (range)	11.38 (6–16)
Mean BMI (range)	25.18 (14.70–33.4)
Number of BMI percentage for age	
Underweight	0 (0%)
Normal weight	14 (18.9%)
Overweight	10 (13.5%)
Obese	50 (67.6%)
Number of slip severity (hips)	
Mild	45 (48)
Moderate	23 (23)
Severe	3 (3)
Number of symptom duration (hips)	
Acute	30 (31)
Chronic	31 (35)
Acute on chronic	7 (8)
Number of Stability (hips)	
Stable	56 (61)
Unstable	12 (13)

**Table 2 jpm-13-00604-t002:** Categorizing results based on the accepted BMI percentile ranges.

	Normal Weight	Overweight	Obese	*p*-Value
Mean age	13.93	11.1	10.72	<0.001
Mean BMI	20.41	23.21	26.91	
Slip severity (Percentage)				0.099
Mild	8 (57.1%)	7 (70.0%)	33 (66.6%)	
Moderate	4 (28.6%)	2 (20.0%)	17 (34.0%)	
Severe	2 (14.3%)	1 (10.0%)	0 (0.0%)	

**Table 3 jpm-13-00604-t003:** Univariate analysis affecting slip severity.

	OR (95% CI)	*p*-Value
Age	0.97 (0.75–1.26)	0.841
Sex		
Male	1 (Reference)	
Female	0.36 (0.09–1.42)	0.147
BMI		
Normal weight	1 (Reference)	
Overweight	0.57 (0.08–4.13)	0.579
Obesity	1.03 (0.27–3.92)	0.965
Symptom duration		
Acute	1 (Reference)	
Chronic	1.05 (0.35–3.18)	0.931
Acute on chronic	2.33 (0.54–10.10)	0.257
Stability		
Stable	1 (Reference)	
Unstable	2.35 (0.66–8.34)	0.185

## Data Availability

Not available.

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
