# Peer review of "Relationship between Slip Severity and BMI in Patients with Slipped Capital Femoral Epiphysis Treated with In Situ Screw Fixation"

_jpm, 2023, doi:10.3390/jpm13040604_

Round 1

Reviewer 1 Report

Dear Author,

Thank you for your paper. The study is quite interesting. There are some points that I could not understand. these are listed below.

1- Why didn't you include the bilateral cases? As 25 % of these patients are bilaterally affected.

2-Why didn't  you include the patients who have endocrine pathology?

3- 68 patients (74 hips). The number of bilaterally affected patients is lower than 25 %. This is not similar previous studies as  reported 25-50 % of all cases.

Kind regards.

Author Response

    • Revision I

    We thank you and the reviewers for your thoughtful suggestions and insights. The manuscript has benefited from these insightful suggestions.

    I look forward to working with you and the reviewers to move this manuscript closer to publication in the Journal of Personalized Medicine

    The manuscript has been rechecked and the necessary changes have been made in accordance with the reviewers’ suggestions. The responses to all comments have been prepared and attached herewith.

    We highlighted the revised part in yellow, and according to the journal's word limit (4000 characters) policy, we added the content in light blue text.

    Thank you for your consideration. I look forward to hearing from you.

    1. Why didn't you include the bilateral cases? As 25 % of these patients are bilaterally affected.
    Author response: Thank you for your comment. We considered a total of 68 patients (74 hips) with SCFE according to inclusion among patients treated with in situ screw fixation. A total of 6 patients with bilateral SCFE were included in this study.

    1. Why didn't you include the patients who have endocrine pathology?
    Author response: Thank you for your review. We aimed to study the relationship between BMI and slip severity in patients with idiopathic SCFE. We excluded patients with endocrine pathology from the inclusion criteria because they were difficult to categorize into idiopathic SCFE.Other studies have separated these conditions similarly; for example, Loader et al. suggested that all patients newly diagnosed with SCFE should be evaluated for the underlying etiology in order to differentiate between idiopathic and atypical SCFE [1]; in addition, patients with obvious underlying disorders (e.g., endocrinopathy, renal failure, radiation therapy etc.) were classified as atypical SCFE [2].

    1. Loder, R.T.; Starnes, T.; Dikos, G. Atypical and typical (idiopathic) slipped capital femoral epiphysis: reconfirmation of the age-weight test and description of the height and age-height tests. JBJS 2006, 88, 1574-1581
    2. Loder R.T.; Skopelja E.N. The epidemiology and demographics of slipped capital femoral epiphysis. International Scholarly Research Notices 2011, 2011:486512

    1. 68 patients (74 hips). The number of bilaterally affected patients is lower than 25 %. This is not similar previous studies as reported 25-50 % of all cases.
    Author response: Thank you for highlighting this. As you mentioned, previous studies reported a higher percentage of patients with bilateral SCFE than our study. We believe that this difference is due to our small sample size; we have added discussion of this into the Discussion:Line 283~ 287: However, our study noted a different proportion of patients with bilateral SCFE compared to other studies. Loader et al. reported that among patients newly diagnosed with SCFE, between 18–50% were diagnosed with bilateral SCFE [31]; our study identified only 13% of patients with the same condition. This difference may be due to the small sample size of our single-center study.

Reviewer 2 Report

Review report

Manuscript ID jpm-2235868

 “Relationship between Slip Severity and BMI in Patients with Slipped Capital Femoral Epiphysis Treated with In Situ Screw Fixation “

Brief summary:

This article is concise and over-all well written. The subject, as the authors note, has to a certain degree been scientifically visited before. The overall results of the present study are inconclusive. Nonetheless the article provides additional information on a relevant subject and “negative” results are, incorrectly, often neglected for publication. The inclusion process, however, needs to be clarified and there are some issues with how the statistical analysis is presented and interpreted. The two methods to measure slip angle should not be  grouped together in statistical analysis since the respective methods provide different information.

Article:
This is a retrospective study of 74 hips with SCFE fixed by in situ fixation.

Abstract:

Line 44. See comment for line 274 further down regarding this conclusion statement left without context.

Introduction:
The introduction is clear and provides relevant background.

Materials and Methods:

• One issue that needs to be clarified is the number of cases (during the same time period) that were not treated by in situ fixation. Where there many cases of reposition and fixation? Where there cases with sub-capitular osteotomies (e.g Dunn procedure)? The presented material cannot be correctly interpreted without this information due to potential selection bias.

• Another major issue of the statistical analysis, as I can interpret it, is the correlation between BMI and slip severity across this age group. The included children range from 6-16 y.o. A BMI of 20 means a very different thing for a 6 y.o and a 16 y.o. The authors correctly use BMI percentiles in figure 2 and sometimes in the text. But throughout the text, as well as in the graphs, I cannot always decipher when the BMI percentile is used for analysis and when BMI is used. The latter is of little value scientifically, with such a range of age.  

• Regarding the measurement of the slip angle there is a rather fundamental risk for misinterpretation of the results since two different methods are used to measure the slip angle. This is unreliable when using categorical classification (e.g mild, moderate) but even more so when using continuous (i.e the Southwick and Wilson angles are not comparable in absolute numbers).

•The Southwick angle can be measured as an absolute value (not the difference compared to the non-affected side) with reliable reference values. Using this method would enable the authors to employ one measurement method instead of two.

• In Figure 2 the BMI is correlated to slip angle. See point above for relevance.

Line 74 There is no mention of the diameter of the screw or if it was fully or partially threaded.

Line 85. One patient was 6 y.o. It is almost unheard of (to my knowledge) of an idiopathic SCFE in this age group. More information should be provided regarding this case.

Line 93-96. Firstly, the reference (20) is not for the CDC categories mentioned in the text. Secondly the BMI categories used (from CDC) could potentially be included as an appendix. Most readers will not know which percentile a BMI of 18 is for, e.g a 14y.o female.

Statistics: It makes sense to present the range rather than the SD for this group. Could the authors clarify why they are using the mean instead of the median for all continuous variables?

Line 133 “The relationship between slip severity and BMI was analyzed using Pearson correlation coefficients”. This analysis provides little information for reasons outlined above.

Discussion:

The issue of selection bias (see M&M) needs to be addressed.

Line 224 and following paragraph. The subgroup consists of only 23 patients. Any statistical analysis of this group runs a high risk of not being reliable. This should be addressed.

Conclusion:

Line 274. The statement regarding the subgroup analysis (moderate slip) must be put into context regarding the small sample size.

Author Response

Revision II

We thank you and the reviewers for your thoughtful suggestions and insights. The manuscript has benefited from these insightful suggestions.

I look forward to working with you and the reviewers to move this manuscript closer to publication in the Journal of Personalized Medicine

The manuscript has been rechecked and the necessary changes have been made in accordance with the reviewers’ suggestions. The responses to all comments have been prepared and attached herewith.

We highlighted the revised part in yellow, and according to the journal's word limit (4000 characters) policy, we added the content in light blue text.

Thank you for your consideration. I look forward to hearing from you.

  1. Article : This is a retrospective study of 74 hips with SCFE fixed by in situ fixation.

  1. Abstract :

1) Line 44. See comment for line 274 further down regarding this conclusion statement left without context.

Author response: Thank you for your comment. We have corrected the abstract and added further context.

  1. Introduction :

1) The introduction is clear and provides relevant background.

  1. Materials and Methods:

1) One issue that needs to be clarified is the number of cases (during the same time period) that were not treated by in situ fixation. Where there many cases of reposition and fixation? Where there cases with sub-capitular osteotomies (e.g Dunn procedure)? The presented material cannot be correctly interpreted without this information due to potential selection bias.

Author response: Thank you for your comment. We performed in situ screw fixation in all patients diagnosed with SCFE and performed additional corrective osteotomy if there was residual deformity in the hip joint after fusion. We have additionally described this issue in the discussion session about potential selection bias: Line 322-323: Third, in situ screw fixation was performed for all patients with SCFE regardless of the initial slip severity. This may have introduced selection bias. 

2) Another major issue of the statistical analysis, as I can interpret it, is the correlation between BMI and slip severity across this age group. The included children range from 6-16 y.o. A BMI of 20 means a very different thing for a 6 y.o and a 16 y.o. The authors correctly use BMI percentiles in figure 2 and sometimes in the text. But throughout the text, as well as in the graphs, I cannot always decipher when the BMI percentile is used for analysis and when BMI is used. The latter is of little value scientifically, with such a range of age.

Author response: Thank you for your comment. As you mentioned, since BMI varies by age in pediatric patients, it is more appropriate to use BMI percentile for age. We modified the statistical analysis by using BMI percentile for age instead of BMI in the correlation analysis. Thank you for helping us to improve our analysis.

3) Regarding the measurement of the slip angle there is a rather fundamental risk for misinterpretation of the results since two different methods are used to measure the slip angle. This is unreliable when using categorical classification (e.g mild, moderate) but even more so when using continuous (i.e the Southwick and Wilson angles are not comparable in absolute numbers).

 4) The Southwick angle can be measured as an absolute value (not the difference compared to the non-affected side) with reliable reference values. Using this method would enable the authors to employ one measurement method instead of two.

Author response: Thank you for your comment. On reflection, we agree that using different methods can lead to misinterpretation of the slip angle results. Therefore, we modified our analysis to use only the Southwick angle to measure the slip angle. In the case of bilateral SCFE, slip severity was classified using 12 degrees as the control angle according to the reference. However, there was no change in the slip severity group when the slip angle was measured using the Southwick angle only in our study group. We have corrected this issue in detail in the material and method session. Line 107~110: For patients with bilateral SCFE, we determined slip severity by measuring the Southwick angle, which uses the control angle of 12 degrees as a reference point [28]. (Figure 1B). Two authors (HS, GRK) who were not involved in the clinical care of the patients participated in the radiographic image measurements. We also changed Figure 1B to show this correction. Line 158~164:Figure 1. Determination of slip severity. (A) Measurement of the Southwick angle difference on a frog-leg lateral radiograph in unilateral SCFE. The slip angle difference was calculated by subtracting b from a; in the example above, the difference was between 30 and 50 degrees, which was classified as moderate slip [27]. (B) Measurement of the Southwick angle in bilateral SCFE on a frog-leg lateral radiograph, which uses 12 degrees as the control angle [28]. The slip angle difference was calculated by subtracting c from d; in the example above, both side were less than 30 degrees, which was classified as mild slip..

5) In Figure 2 the BMI is correlated to slip angle. See point above for relevance.

Author response: Thank you for your comment. As previously discussed, we modified the statistical analysis to use BMI percentile for age instead of BMI in the correlation analysis. We have updated Figure 2 to reflect this change.
6) Line 74 There is no mention of the diameter of the screw or if it was fully or partially threaded.

Author response: Thank you for your suggestion. We have added this detail in the material and methods session.

Line 95~109: We performed all surgeries with in situ screw fixation regardless slip severity. Under general anesthesia, the affected leg was held in extension and slight internal rotation position on the traction table in supine position, and hemi-lithotomy position was performed on unaffected leg. After positioning the patient, we checked whether the true anterior to posterior (AP) hip radiograph and lateral radiograph of the affected leg could be checked with the C-arm. Under fluoroscopic guidance, a cannulated screw guidewire was inserted percutaneously in the center-center position of the deformed femoral epiphysis. The guidewire was inserted up to about 3 mm from the femoral articular surface. After making a small stab incision around the guidewire, bone was drilled. A 6.5 or 7.3 mm one partial threaded cannulated screw was inserted over the guidewire after reaming. After cannulated screw insertion, internal & external rotation was carefully performed on the traction leg to confirm that the screw had not penetrated into the articular surface. The guidewire was removed and stab wound was closed after confirming the satisfactory screw position. Patients were allowed to protected partial weight bearing with crutches for 6 weeks. We checked serial hip radiographs at 3 months, 6 months, and 1 year to check for any complications on the affected side and newly detected slip on the unaffected side.

7) Line 85. One patient was 6 y.o. It is almost unheard of (to my knowledge) of an idiopathic SCFE in this age group. More information should be provided regarding this case.

Author response: Thank you for your comment. Although there are some differences in the literature, the average age of onset of SCFE is 12 years for boys and 11 years for girls and other authors have reported that the typical age for the onset of SCFE is 8 to 15 years. As you mentioned, diagnosis with idiopathic SCFE at the age 6 is unusual. In our institution, hormonal tests such as TSH, free T4 and growth hormone are performed routinely to check for endocrine dysfunction in SCFE patients with development at an atypical age. In the discussed patient, no endocrine abnormality was found. We have additionally described this issue in detail in the discussion session. Loader et al. reported that the underlying etiology should be evaluated in all patients newly diagnosed with SCFE in order to differentiate between idiopathic and atypical SCFE. Burrow et al. also reported that patients below the tenth percentile for chronological age should be evaluated for the possibility of endocrine dysfunction. Line 288~292: In our study group, the average age of patients was 11.4 years. Loader et al. reported that the average age of SCFE onset is 12 years for males and 11 years for females, and that the underlying etiology should be evaluated to differentiate idiopathic SCFE from atypical SCFE [32]. In our study, we routinely performed hormonal tests on all patients diagnosed with SCFE, and classified patients with negative results as idiopathic SCFE.

8) Line 93-96. Firstly, the reference (20) is not for the CDC categories mentioned in the text. Secondly the BMI categories used (from CDC) could potentially be included as an appendix. Most readers will not know which percentile a BMI of 18 is for, e.g a 14y.o female.

Author response: Thank you for your review. We have corrected this error in referencing.

9) Statistics: It makes sense to present the range rather than the SD for this group. Could the authors clarify why they are using the mean instead of the median for all continuous variables?

Author response: Thank you for your comment. In this data, age and BMI both satisfied the normality test and there was no difference between the mean and median values. We present the mean because it satisfied the normality test. As per your suggestion, we have presented the ranges for the means rather than the standard deviations.

10) Line 133 “The relationship between slip severity and BMI was analyzed using Pearson correlation coefficients”. This analysis provides little information for reasons outlined above.

Author response: Thank you for your comment. We modified the statistical analysis by using BMI percentile for age instead of BMI in the correlation analysis. With the new statistical analysis method, we similarly found no correlation between all groups

Discussion:

11) The issue of selection bias (see M&M) needs to be addressed.

Author response: Thank you for your comment. We initially performed in situ screw fixation in all patients diagnosed with SCFE and performed additional corrective osteotomy if there was residual deformity in the hip joint after fusion. We have additionally described this issue in detail in the discussion session.

Line 322-325: Third, in situ screw fixation was performed for all patients with SCFE regardless of the initial slip severity. This may have introduced selection bias. In particular, only in situ screw fixation was used in severe SCFE patients. The patients with severe SCFE were all chronic SCFE. After sufficiently discussing the treatment with the patient and parents, in-site screw fixation was decided.

12) 224 and following paragraph. The subgroup consists of only 23 patients. Any statistical analysis of this group runs a high risk of not being reliable. This should be addressed.

Author response: Thank you for your comment. As you mentioned, the small sample size in the subgroup analysis may have had a negative impact on the statistical power of the study. We had added discussion of this as a limitation.

Conclusion:

13) Line 274. The statement regarding the subgroup analysis (moderate slip) must be put into context regarding the small sample size.

Author response: Thank you for your comment. As mentioned, we modified the statistical analysis method. Regardless of sample size, we did not find any association in subgroup analysis. However, we have added discussion of the small sample size to our manuscript. Thank you for helping us to improve the manuscript.

Round 2

Reviewer 2 Report

The authors have presented reasonable revisions based on the review.